# Apolipoprotein Signature of HDL and LDL from Atherosclerotic Patients in Relation with Carotid Plaque Typology: A Preliminary Report

**DOI:** 10.3390/biomedicines9091156

**Published:** 2021-09-03

**Authors:** Francesco Finamore, Gabriele Nieddu, Silvia Rocchiccioli, Rita Spirito, Anna Guarino, Marilena Formato, Antonio Junior Lepedda

**Affiliations:** 1Institute of Clinical Physiology, National Research Council, via Giuseppe Moruzzi 1, 56124 Pisa, Italy; f.finamore@fpscience.it (F.F.); silvia.rocchiccioli@ifc.cnr.it (S.R.); 2Department of Biomedical Sciences, University of Sassari, Viale San Pietro 43, 07100 Sassari, Italy; ganieddu@uniss.it (G.N.); formato@uniss.it (M.F.); 3Centro Cardiologico Monzino, IRCCS, via Parea 4, 20138 Milano, Italy; ritaspirito@virgilio.it (R.S.); anna.guarino@ccfm.it (A.G.)

**Keywords:** lipoproteomics, atherosclerosis, plaque instability

## Abstract

In the past years, it has become increasingly clear that the protein cargo of the different lipoprotein classes is largely responsible for carrying out their various functions, also in relation to pathological conditions, including atherosclerosis. Accordingly, detailed information about their apolipoprotein composition and structure may contribute to the revelation of their role in atherogenesis and the understanding of the mechanisms that lead to atherosclerotic degeneration and toward vulnerable plaque formation. With this aim, shotgun proteomics was applied to identify the apolipoprotein signatures of both high-density and low-density lipoproteins (HDL and LDL) plasma fractions purified from healthy volunteers and atherosclerotic patients with different plaque typologies who underwent carotid endarterectomy. By this approach, two proteins with potential implications in inflammatory, immune, and hemostatic pathways, namely, integrin beta-2 (P05107) and secretoglobin family 3A member 2 (Q96PL1), have been confirmed to belong to the HDL proteome. Similarly, the list of LDL-associated proteins has been enriched with 21 proteins involved in complement and coagulation cascades and the acute-phase response, which potentially double the protein species of LDL cargo. Moreover, differential expression analysis has shown protein signatures specific for patients with “hard” or “soft” plaques.

## 1. Introduction

Plaque rupture/erosion leading to atherothrombosis is the underlying pathology responsible for major acute events in cardiovascular disease, such as stroke, acute coronary syndrome, and peripheral artery occlusion [1,2]. To date, many efforts have been dedicated to elucidating the determinants of carotid plaque vulnerability and identifying reliable and specific markers for the susceptibility of plaques to rupture and erosion [3], including vascular imaging [4]. In the past twenty years, with the improvement of proteomic tools, large-scale technologies have been applied to elucidate pathways of atherosclerotic degeneration and identify new circulating markers to be utilized either as early diagnostic traits or as targets for new drug therapies [5]. To address these issues, several proteomic approaches, such as 1D- and 2D-electrophoresis, followed by mass spectrometry (MS) analyses, protein arrays, and gel-free MS-based proteomics, have been applied to different matrices such as vascular cell/tissues, looking at both proteomes and secretomes [6,7], plasma/serum [8], urine [9], and purified plasma lipoprotein fractions [10,11]. As far as our research group is concerned, we have been focusing on understanding the mechanisms that lead to atherosclerotic degeneration and toward stable or unstable advanced lesions, looking for a proteomic signature of vulnerable plaque [12,13], including in relation to post-translational oxidative modifications [14,15].

Lipoproteins, especially low-density and high-density lipoproteins (LDL and HDL), have attracted a great deal of interest because of their implication in atherogenesis. LDL, the main carrier of cholesterol to the peripheral tissues, is a well-established major risk factor for atherosclerosis and cardiovascular disease [2]. In fact, the selective retention of LDL into the subendothelial space by means of specific interactions with extracellular matrix proteoglycans is thought to be the initiating event during atherogenesis [16]. HDL has major vascular protective effects, being responsible for the removal of cholesterol excess from cells and its transport to the liver (reverse cholesterol transport) [17], as well as for its antioxidant, anti-inflammatory, and anti-thrombotic properties [18]. Furthermore, Vaisar T et al., by applying Shotgun proteomics, revealed that HDL is enriched in several proteins involved in the complement cascade, suggesting a role in innate immunity [19]. Indeed, it is becoming increasingly clear that the HDL protein cargo, rather than HDL-cholesterol levels, is crucial for its vascular function, even turning HDL from anti-atherogenic to pro-atherogenic in cardiovascular disease [19,20,21,22]. In this respect, we have evidence of increased levels of acute-phase serum amyloid A protein in HDL as well as VLDL and LDL plasma fractions purified from patients undergoing carotid endarterectomy [23]. Interestingly, it was shown that coronary artery disease (CAD) risk-lowering therapy with statin/niacin partially reverts the changes in HDL proteome observed in CAD subjects [24].

Since the protein cargo of these particles is largely responsible for carrying out their various functions, detailed information about the apolipoprotein composition and structure may contribute to revealing their role in atherogenesis and developing new therapeutic strategies for the treatment of lipoprotein-associated disorders. Accordingly, in the last seventeen years, a plethora of proteomic studies have been performed on purified lipoprotein fractions in both physiological [19,25] and pathological conditions, including CAD [26,27], acute myocardial infarction [28], chronic kidney disease [29,30], end-stage renal disease [31], type 1 and 2 diabetes mellitus [32,33,34], and experimental atherosclerosis [35,36] (see Appendix A). An updated list of all identified HDL and LDL proteins, ranked according to the frequency of identification in MS studies, can be found at the “The Davidson/Shah lab” website www.DavidsonLab.com (last update 24 April 2019).

The aim of this study is to identify an apolipoprotein signature of both HDL and LDL plasma fractions from atherosclerotic patients who underwent carotid endarterectomy in relation to plaque typology, to be compared with data obtained from a normolipidemic control group. To our knowledge, this is the first shotgun proteomic study on the association between plasma apolipoprotein composition and carotid plaque echogenicity.

## 2. Materials and Methods

### 2.1. Sample Collection

HDL and LDL fractions were purified from pooled plasma samples from 75 patients undergoing carotid endarterectomy at Centro Cardiologico Monzino, IRCCS (Milano) and 50 healthy normolipidemic volunteers matched for age and sex with the patients, enrolled in previous published studies [13,23]. All patients underwent surgery according to NASCET guidelines for carotid stenosis [37]. Carotid atherosclerosis was assessed by ultrasonography using a Mylab 70 X Vision Echocolor Doppler instrument equipped with an LA332 AppleProbe 11–3 MHz (Esaote, Genova, Italy). All patients selected for surgery had either high-grade stenosis (>70%) or an ulcerated lesion of a medium grade based on Echo-Doppler analysis. Plaques were classified according to Gray-Weale classification [38] into five types: uniformly echolucent plaque (type 1), predominantly echolucent plaque with <50% echogenic areas (type 2), predominantly echogenic plaque with <50% echolucent areas (type 3), uniformly echogenic plaque (type 4), and plaque with heavy calcification and acoustic shadows (type 5). This method represents an effective and non-invasive way to detect and characterize atherosclerotic plaque in peripheral arteries and provides a rapid risk stratification of the patient through the Gray-Weale standardization [39,40,41,42]. Patients were sorted, according to the plaque typology, into “soft” (*n* = 44), having a carotid plaque with hypoechoic features (types 1 and 2), and “hard” (*n* = 31), having a carotid plaque with hyperechoic features (types 3, 4, and 5). In this respect, there is plenty of evidence that carotid plaque echolucency provides predictive information in asymptomatic carotid artery stenosis, as reviewed in the meta-analysis study by Gupta et al. [43]. The main clinical parameters of the two groups of patients are summarized in Table 1. Before surgery, fasting blood samples were collected into Vacutainer tubes containing EDTA, immediately centrifuged at 2000× *g* for 10 min at 4 °C, and stored at −80 °C until analysis. Informed consent was obtained before enrolment. Institutional Review Board approval was obtained. The study was conducted in accordance with the ethical principles of the current Declaration of Helsinki.

All reagents and chemicals used in this study were of analytical grade from Merck-Millipore (Darmstadt, Germany), unless otherwise stated.

### 2.2. Lipoproteins Purification

Lipoproteins were isolated by isopycnic salt gradient ultracentrifugation, as previously reported [23]. Briefly, 3.9 mL of pooled plasma samples were adjusted with solid NaBr to *d* = 1.3 g/mL (472.2 mg NaBr/mL plasma) in Thin Wall Ultra-Clear centrifuge tubes (Beckman Coulter, Indianapolis, IND, United States), gently overlaid with 8.1 mL of a 0.6% NaCl solution (*d* = 1.006 g/mL), and centrifuged at 285,000× *g* for 48 h at 4 °C in an Optima L90 series ultracentrifuge instrument equipped with a SW40 Ti rotor (Beckman Coulter, Indianapolis, IND, United States). Afterwards, both LDL (*d* = 1.063–1.19 g/mL) and HDL (*d* = 1.19–1.21 g/mL) fractions were collected and further purified by a second centrifugation step, performed at 285,000× *g* for 24 h in saline solutions at density 1.063 and 1.21 g/mL, respectively. Finally, both fractions were desalted to a final salt concentration < 5 mM and concentrated using Amicon Ultra-0.5 mL centrifugal filter units (10 KDa MWCO, Merck-Millipore, Darmstadt, Germany). Protein concentration was determined using the DC Protein Assay Kit (Bio-Rad Laboratories, Hercules, CA, USA), according to the manufacturer’s instructions, using bovine serum albumin as standard.

Hereafter, for the sake of simplicity, lipoprotein fractions will be called HDL “hard” and LDL “hard” as well as HDL “soft” and LDL “soft”, according to the plaque typology. Moreover, fractions purified from pooled plasma from healthy controls will be called HDL “CNTR” and LDL “CNTR”.

### 2.3. SDS-PAGE Analysis

The degree of purity of both LDL and HDL fractions was evaluated by mono-dimensional electrophoresis on polyacrylamide gels (PAGE). Samples were diluted with 4X Laemmli buffer, consisting of 250 mM Tris, pH 6.8, 8% sodium dodecyl sulfate (SDS) (*w/v*), 8% dithiothreitol (DTT) (*w*/*v*), 40% glycerol (*v*/*v*), and 0.0008% bromophenol blue (*w/v*), boiled for 5 min, and resolved by Tris-glycine SDS-PAGE. For LDL, 4% T and 3% C stacking gel, 5% T and 3% C running gels were cast, whereas HDL apolipoproteins profile was obtained using 15% T running gels. Electrophoresis was carried out at 50 V for the first 15 min and then at 150 V until the bromophenol dye front reached the lower limit of the gel in a Mini-Protean II cell vertical slab gel electrophoresis apparatus (Bio-Rad Laboratories, Hercules, CA, USA). Following Coomassie brilliant blue G-250 staining, gels were acquired at 63 μm resolution using a GS-800 calibrated densitometer (Bio-Rad Laboratories, Hercules, CA, USA).

### 2.4. Shotgun Proteomics

Shotgun proteomics was performed on HDL and LDL fractions purified from pooled plasma samples from 31 patients with hard plaque, 44 patients with soft plaque, and 50 controls, as previously described, with some modifications [44]. First, 50 µg of proteins from each fraction were diluted with 100 mM NH_4_HCO_3_ with 1% sodium deoxycholate (SDC) to improve protein unfolding and subsequent endoprotease digestion. Cysteine residues were reduced with 10 mM DTT at 65 °C for 30 min, followed by their alkylation with 22 mM iodoacetamide at room temperature for 1 h in dark conditions. Proteins were digested overnight at 37 °C using trypsin (1:50 *w*/*w*). Acidic precipitation of SDC was performed by incubating each sample with 1% trifluoroacetic acid at 37 °C for 45 min. Samples were centrifuged at 16,000× *g* for 10 min and subsequently desalted with Mobicol spin columns (MoBiTec Molecular Biotechnology, Goettingen, Germany) equipped with 10 µm pore size filters and filled with VersaFlash C18 spherical 70 Å silica particles. Peptides were dried under vacuum and subsequently dissolved in 0.1% formic acid to achieve an estimated peptide concentration of 0.5 µg/µL prior to liquid chromatography tandem mass spectrometry (LC-MS/MS) data acquisition. Samples were injected three times to assess the technical variability of the instrument. The peptides were analyzed by nanoLC-MS/MS on an EASY-nLC coupled with an Orbitrap Fusion Tribrid mass spectrometer equipped with an EASY spray source (Thermo Fisher Scientific, Waltham, MA, USA). The peptides were trapped on a PepMap C18 precolumn (2 cm × 75 µm, i.d., 3 µm particle size, 100 Å pore size) and separated at a flow rate of 300 nL/min onto a PepMap C18 column of 50 cm × 75 µm i.d., 2 µm, 100 Å pore size heated at 35 °C. The analytical separation was run using a 125 min gradient of 0.1% formic acid (mobile phase A) and acetonitrile/0.1% formic acid (mobile phase B) as follows: from 5% to 22% B in 104 min, from 22% to 32% B in 15 min, from 32% to 90% B in 10 min. MS data were acquired in positive ion mode with a spray voltage of 2.1 kV and a capillary temperature of 275 °C. A full MS1 survey scan was acquired with a mass range of 375–1200 m/z, resolving power of 120,000 FWHM (at 200 m/z), and a maximum injection time of 50 ms. The most intense precursors (minimum intensity threshold of 5000, 2–7 charge state) were quadrupole-isolated (isolation width of 1.6 m/z) and fragmented by higher energy collisional dissociation (HCD) at the normalized collision energy (NCE) of 27%. Detection was carried out in the dual-pressure ion trap (IT) with an AGC of 2000 and a maximum injection time of 300 ms. A dynamic exclusion of 60 s was enabled. Raw data were converted into mzML files and queried against the human UniprotKB/Swiss-Prot TrEMBL database (188558 sequences) (September 2019) using X!Tandem (TPP tool) [45]. The parameters used to match the detected features included in each peak list and the theoretical masses were as follows: a precursor and fragment ion tolerance of 10 ppm and 0.5 Da, respectively; 2 missed cleavages allowed for trypsin digestion; cysteine residues carbamidomethylation as static modification (+57.021464 Da); methionine oxidation (+15.994915 Da) and N-terminal acetylation (+42.010565 Da) as variable modifications. False discovery rate (FDR) was performed by matching the peak list data against a decoy database composed of reversed protein sequences. The estimated number of false-positive peptide identifications was then calculated to filter the true positive matches according to an FDR ≤ 5% using Mayu v1.06 software [46]. The generated pepXML files were manually processed using the MS1 full-scan filtering option of Skyline software v3.1 (McCoss Lab, Seattle, WA, USA) [47] to extract the peak areas of all the identified peptides. Peak areas values of peptides belonging to the same protein were integrated to obtain total protein abundance per entry. Protein abundances were normalized by the total abundance of all the identified proteins for each run. Proteins were considered to be differentially expressed with an absolute fold change (|FC|) > 3.

### 2.5. Protein–Protein Interaction (PPI) Network and Gene Ontology (GO) Analysis

Functional PPI networks were generated using only the differentially expressed proteins queried against the STRING v11.0 database in Cytoscape v3.8.2 [48] with a cut-off confidence score equal to 0.4. All nodes (proteins) with a number of edges lower than 4 were filtered out from the network construction. Proteins belonging to each network were assessed for biological processes using the Cytoscape plug-in ClueGO [49] using the following parameters: a *p*-value ≤ 0.01 integrated with a Bonferroni step down correction; a GO tree interval between 4 and 8; a minimum number of genes per cluster of 4, with 4% of genes; a kappa score of 0.4; an initial group size of 3 terms, with a percentage of overlapping terms per group of 50%. GO term fusion of similar associated genes was enabled.

## 3. Results

### 3.1. Lipoproteins Purification

LDL and HDL fractions were purified from pooled plasma samples from patients undergoing carotid endarterectomy with either soft or hard plaques and from healthy normolipidemic volunteers. The adopted purification procedure, an isopycnic salt gradient ultracentrifugation followed by a further step of fraction flotation by high centrifugal fields, provided highly purified LDL and HDL fractions, as shown by their 1-D electrophoretic profiles (Figure 1). In particular, none of the two fractions were grossly contaminated by albumin. Furthermore, as expected, LDL fractions were composed of apolipoprotein B100 for more than 95% of LDL apolipoproteins (upper part of the lane, panel A), whereas apolipoprotein AI represented over 80% of HDL apolipoproteins (panel B).

### 3.2. Proteomics Characterization of HDL and LDL

Protein extracts from both HDL and LDL fractions were digested and analyzed using a data-dependent mass method coupled to label-free quantification at the MS1 level. A total of 129 proteins were quantified from the HDL fraction, with an average squared Pearson correlation (R^2^) of 0.973 and an average coefficient of variation (%CV) of 31.7%. Concerning the LDL fraction, 87 proteins were quantified with an average R^2^ of 0.978 and an average %CV of 10.9%. HDL and LDL fractions shared 61 proteins, whereas 68 and 26 were uniquely found in HDL and LDL fractions, respectively (Figure 2, Appendix A).

Identified proteins were compared with a reference database (http://www.DavidsonLab.com, last update 24 April 2019). Among the 129 proteins identified in HDL, 84 were already listed as “likely” HDL proteins, whereas two proteins, previously reported by two other laboratories, namely, integrin beta-2 (ITGB2) and secretoglobin family 3A member 2 (SCGB3A2), potentially increase the number of proteins that could be considered HDL-associated to 221 (Appendix A). With regard to LDL, 20 proteins were already among the “likely” LDL proteins, while 21 proteins, involved in complement and coagulation cascades (Figure 3) [25,50], potentially double the list of protein species associated with high confidence to LDL (Appendix A).

### 3.3. Differential Expression Analysis

Differential expression analysis in both HDL and LDL was performed by comparing protein abundances between fractions purified from “soft”, “hard”, and “CNTR” plasma pools using a |FC| > 3.

A total of 84 and 16 proteins were found to be differentially expressed in HDL and LDL fractions from atherosclerotic patients compared to controls, respectively (Appendix A). Concerning HDL, 37 proteins were found to be down-regulated and 1 up-regulated in the “hard” group, while 6 proteins were found down-regulated and 59 proteins up-regulated in the “soft” group. Moreover, LDL showed 8 down-regulated and 1 up-regulated proteins in the “hard” group, while 4 and 5 proteins were, respectively, under- and over-expressed in “soft” plaques. Differentially expressed proteins were subsequently used to generate PPI networks to study the functional association between them. A highly interconnected network was obtained for HDL “hard” plague (Figure 4A), in which most of the proteins were shown to be involved in biological functions related to protein processes, including maturation (PCSK9, APOH), transport (HBA1, HBB, TTR, TF, LPA), localization (CD44, LPA, F2, PCYOX1), inflammatory response (C4A, C4B, F2, A2M), wound healing (SERPIND1, F2, CD44), and coagulation (SERPIND1, F2, CD44) (Figure 4B). Moreover, the HDL “soft” network (Figure 4C) evidenced a significant enrichment in proteins involved in acute inflammatory response (SERPING1, LBP, ORM1, and C4BPA), defense response (ITGB1, PPBP, and DCD), response to wounding (NOTCH3, ITGB1, ITGB3, ITGA2, and AGT), and lipid transport (LBP, APOL1, LPA, APOH, and APOF) (Figure 4D). Similarly, networks generated from differentially expressed proteins of LDL “hard” (Figure 5A) and “soft” (Figure 5C) plaques showed a high significance of biological processes related to acute phase response and inflammation (ORM2, DCD, CD59, and A2M) for “hard plaques” and lipid metabolic processes, transport, localization, and homeostasis (LRP2, APOH, APOC2, APOC4, and APOF) for “soft plaques” (Figure 5B,D).

By comparing the protein abundances between “soft” and “hard” plaques (soft-to-hard ratio), 82 proteins were found up-regulated and 4 proteins down-regulated in HDL “soft” plaque (Appendix A). With regard to LDL, 15 and 1 proteins were up- and down-regulated in “soft” plaques, respectively. Additionally, for these proteins, functional networks were generated (Figure 6A,C, respectively, for HDL and LDL). GO analyses showed significant enrichment in proteins involved in immune system processes, inflammation, hemostasis, and lipid transport (to cite few) in the HDL “soft” group, whereas terms such as “regulation of lipoprotein particle remodeling”, “regulation of lipoprotein lipase activity”, “lipid metabolic process”, and “lipid transport” were enriched in the LDL “soft” group (Figure 6B,D, respectively, for HDL and LDL).

## 4. Discussion

Carotid atherosclerosis represents a relevant healthcare problem as unstable plaques cause approximately 15% of neurologic events, namely, transient ischemic attack (TIA) and stroke [51], the latter being responsible for 11.6% of global deaths [https://vizhub.healthdata.org/gbd-compare/, last update 24 April 2019].

Several studies have shown that plaque instability is associated with a substantial increase in inflammatory and proteolytic activity [12,52], oxidative modifications [15], lipid accumulation, thrombosis, and angiogenesis [2,51]. Understanding the mechanisms leading to plaque stabilization or to its evolution towards ulceration is mandatory to prevent major adverse clinical events related to thrombosis and artery occlusion. In this respect, the identification of serum biomarkers of vulnerable carotid plaques may also be useful for selecting patients requiring surgery [51].

We have performed a shotgun proteomic analysis on highly purified HDL and LDL fractions from pooled plasma samples obtained through two ultracentrifugation steps in high-salt media from patients who underwent carotid endarterectomy and healthy controls, which allowed for the comparison of levels of tens of proteins simultaneously. Plasma samples from patients were pooled according to carotid plaque echogenicity using ultrasonography, a well-established non-invasive method to assess plaque characteristics and the severity of carotid stenosis [39,40,41,42,43]. The use of pooled samples represents the major limitation of the present study since each piece of data reported represents an average value while a measurement of the level of variation or dispersion from the average is missing.

As reported by several studies, the method used to isolate lipoproteins significantly affects the protein content of the resulting particles. The most widely used methods, established in the 1950s [53], rely on ultracentrifugation in high-salt media containing KBr or NaBr for lipoprotein purification (see Appendix A). Although the high ionic strength, together with the high centrifugal field forces, might cause the loss of some proteins from the lipoprotein particle, these methods allow us to identify proteins that are, with high confidence, actually associated with a given lipoprotein fraction. In some applications, salts may be partially substituted by deuterium oxide, therefore reducing the ionic strength of the solution [54]. Alternatively, lipoproteins can be isolated by immunopurification [19], free solution isotachophoresis [55], or chromatographic techniques, such as fast protein liquid chromatography [56] and size exclusion chromatography [57]. Even though this probably reduces the loss of weakly associated protein, at the same time, it leads to non-specific co-purification of protein contaminants or, in the case of immunopurification, other lipoprotein fractions bearing the same antibody target.

According to “The HDL Proteome Watch Database” by the Davidson Laboratory (http://www.DavidsonLab.com, last update 24 April 2019), which includes 41 proteomics studies on HDL, published up to 2020, 819 proteins have been identified, of which 219 (defined “likely” HDL proteins) have been reported by at least three different laboratories. In this study, 129 proteins were identified, including 84 “likely” HDL proteins and two proteins previously reported by two other laboratories, integrin beta-2 (P05107) and secretoglobin family 3A member 2 (Q96PL1), which potentially increase the number of proteins that could be considered HDL-associated to 221.

With regard to LDL, the Davidson database lists 60 proteins from 4 proteomics studies up to 2015; of these, 22 (defined “likely” LDL proteins) were identified independently by at least two studies. We identified 87 LDL associated proteins, of which 20 were from the “likely” LDL proteins list and 21 were already reported by either Dashty et al. (20 out of 21) [25] or Bancells C et al. (Protein S100-A9, P06702) [50], potentially doubling the protein species of the LDL cargo. Interestingly, GO analysis has shown a functional association between these proteins and complement activation, as already evidenced by the fact that LDL and its acetylated form induce complement activation and C3b opsonization [58].

Commonly identified proteins in both HDL and LDL included 15 apolipoproteins (A-I, A-II, A-IV, A-V, B-100, C-I, C-II, C-III, C-IV, D, E, F, L1, M, (a)), complement factors (C3, C4-A, C4-b, and C9), fibrinogen (α, β, and γ chains), and several immunoglobulin isoforms and serum amyloid proteins (A-1, A-2, and A-4). The 68 proteins uniquely identified in the HDL fraction included proteins involved in innate immune response (C4BPA, SERPING1, CAMP), extracellular matrix organization (CD44, ITGA2, ITGA2B, ITGB3), and acute phase response (F2, ITIH4, LBP, and SERPINF2). On the other hand, proteins involved in “leukocyte migration” (ITGA6, SELL), “platelet degranulation” (CD9, A1BG, TUB4A), and “cell surface signaling pathway” (CD59, CD9, MADD) were exclusively found in the LDL fraction.

Both “hard” and “soft” lipoprotein fractions shared proteins involved in biological processes such as inflammatory and immune responses, complement activation, and blood coagulation, confirming the relevance of these events in the generation and worsening of atherosclerotic plaques, as already shown by a large amount of evidence [59,60,61,62].

Particularly, some proteins that might contribute to increase cardiovascular risk in patients with soft plaque showed higher expression levels in HDL “soft” compared to “hard”, including complement factors C3 and C4, which have been described to promote atherosclerosis [63]; integrins, which are a class of proteins that may mediate the recruitment of inflammatory cells to atherosclerotic plaques [64]; prenylcysteine oxidase 1 [65], which is a pro-oxidant enzyme; transthyretin [66]; and lipopolysaccharide-binding protein, which has been previously evidenced to play a role in atherosclerosis development [67] and cardiovascular mortality [68]. Moreover, some SAA proteins (SAA1, SAA2, and SAA4) were found up-regulated in the HDL “soft” group but down-regulated, with the exception of SAA1, in the LDL “soft” group. Despite the role of SAA proteins in atherogenesis still being unclear, evidence has shown that their altered regulation is correlated with an increased cardiovascular risk through the binding of lipoproteins to vascular extracellular matrix proteoglycans [69,70] and the favoring of chemiotaxis by activating toll-like receptors via serum amyloid A [71].

## 5. Conclusions

By applying shotgun proteomics to the analysis of lipoproteomes, this study has provided new insights into the protein composition of both HDL and LDL, widening the already substantial HDL proteome with two additional HDL-like proteins and the LDL proteome with twenty-one additional LDL-like proteins, with potential implications in inflammatory, immune, and coagulation pathways. Furthermore, differential expression analysis has shown protein signatures specific for patients with “hard” or “soft” plaques, which represent a useful starting point for further research.

## Figures and Tables

**Figure 1 biomedicines-09-01156-f001:**
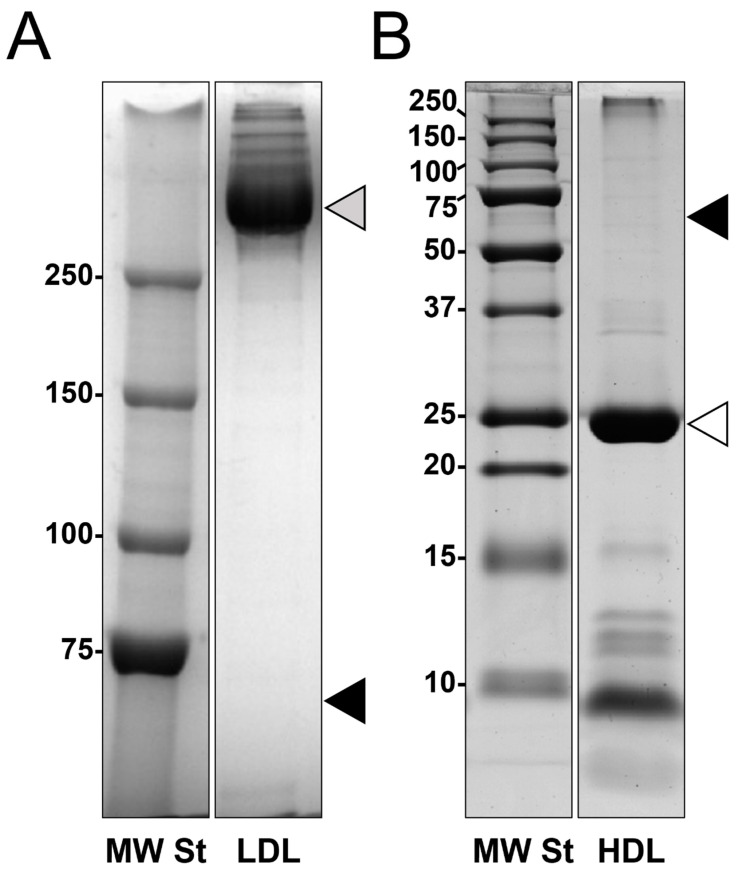
Representative mono-dimensional profiles of both LDL (panel **A**) and HDL (panel **B**) fractions purified by isopycnic salt gradient ultracentrifugation followed by a further ultracentrifugation step. Fractions were desalted and concentrated using centrifugal filter units with 10 KDa cut-off and resolved by SDS-PAGE in 5% T (panel **A**) or 15% T (panel **B**) gels. Grey arrowhead indicates the apolipoprotein B100 band (>550 kDa MW) (panel **A**), and white arrowhead indicates the apolipoprotein AI band (≃28 kDa MW) (panel **B**). In both panels, the black arrowheads indicate where the albumin (≃66 kDa MW) band would be located.

**Figure 2 biomedicines-09-01156-f002:**
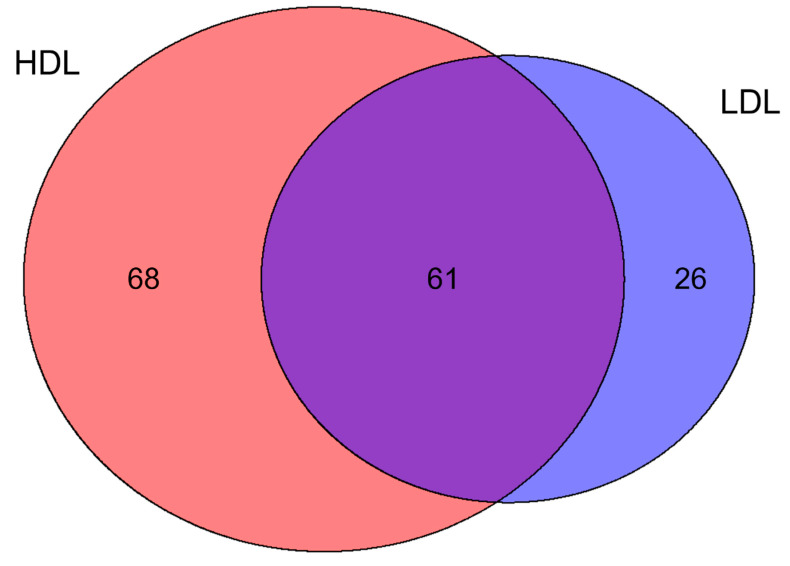
Venn diagram showing the distribution between HDL and LDL of the 155 proteins identified. The two lipoprotein fractions shared 61 proteins, whereas 68 and 26 proteins were exclusive to HDL and LDL, respectively.

**Figure 3 biomedicines-09-01156-f003:**
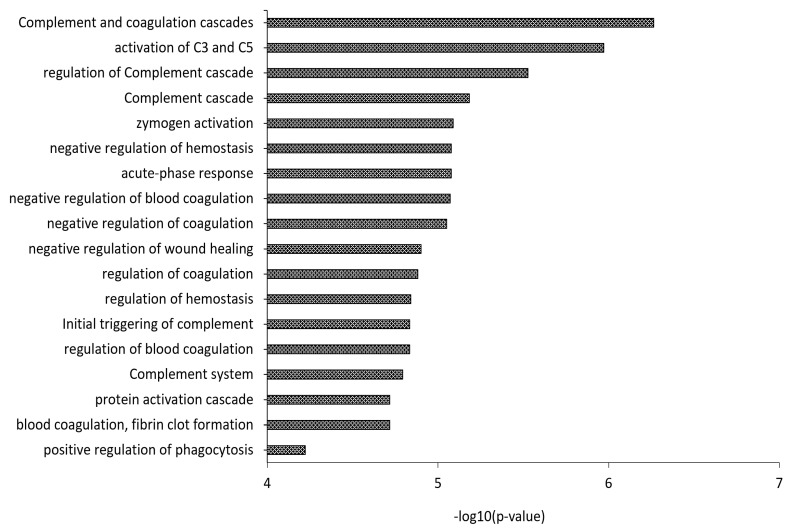
Gene Ontology (GO) analysis of the 21 proteins identified in this study, which are not yet included in the “likely” LDL proteins list of the Davidson’s Lab database (http://www.DavidsonLab.com, last update 24 April 2019), already reported by either Dashty et al. [25] or Bancells et al. [50] to be associated with LDL.

**Figure 4 biomedicines-09-01156-f004:**
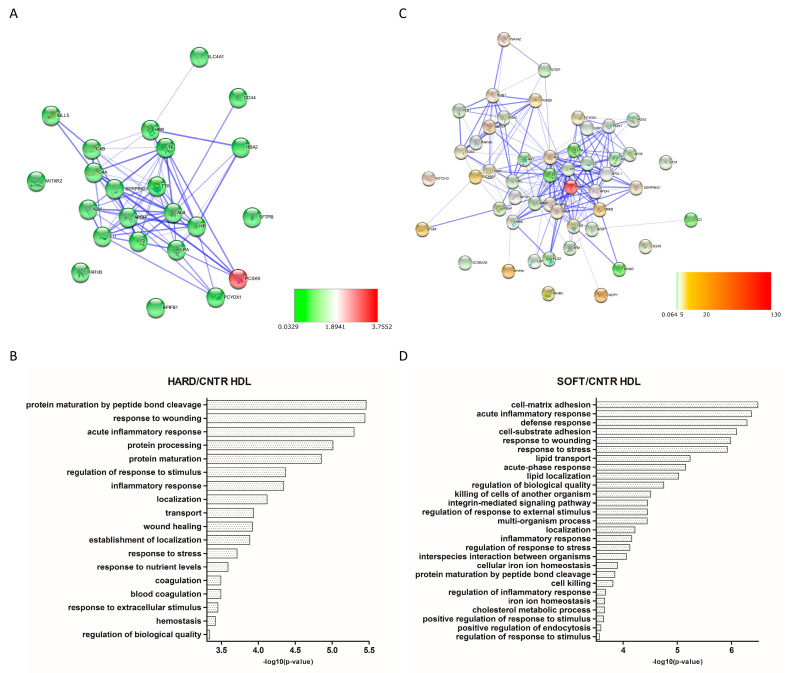
Protein–protein interaction (PPI) network and Gene Ontology (GO) analysis of differentially expressed proteins in HDL “hard” plaques (**A**,**B**, respectively) and HDL “soft” plaques (**C**,**D**, respectively), showing the biological processes in which these proteins are involved. Colors of nodes in PPI networks are representative of FCs compared to controls.

**Figure 5 biomedicines-09-01156-f005:**
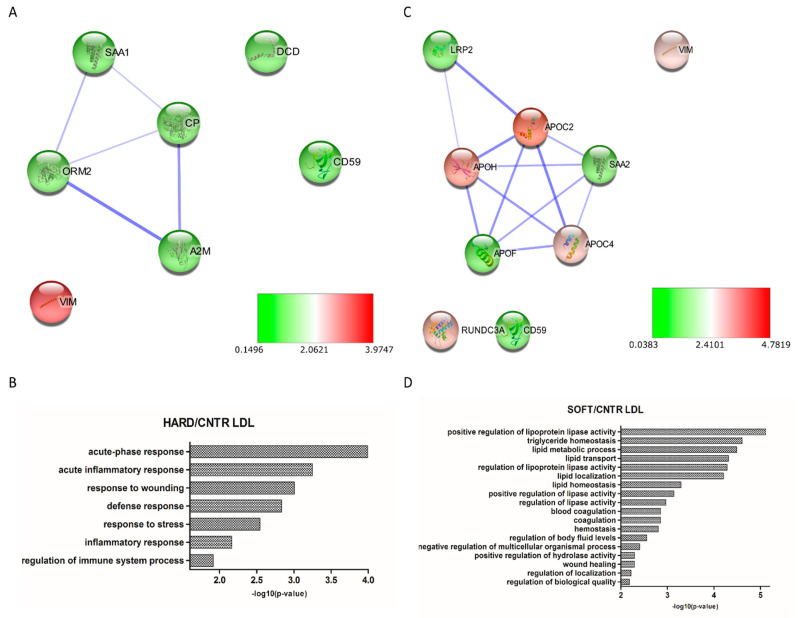
Protein–protein interaction (PPI) network and Gene Ontology (GO) analysis of differentially expressed proteins in LDL “hard” plaques (**A**,**B**, respectively) and LDL “soft” plaques (**C**,**D**, respectively), showing the biological processes in which these proteins are involved. Colors of nodes in PPI networks are representative of FCs compared to controls.

**Figure 6 biomedicines-09-01156-f006:**
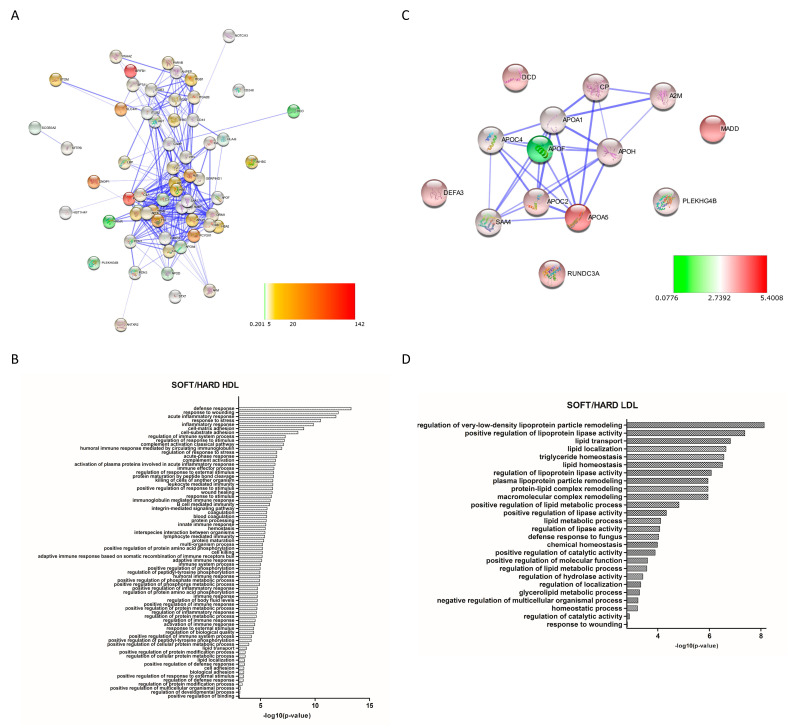
Protein–protein interaction (PPI) network and Gene Ontology (GO) analysis of differentially expressed proteins between HDL “soft” and “hard” plaques (**A**,**B**, respectively) and LDL “soft” and “hard” plaques (**C**,**D**, respectively), showing the biological processes in which these proteins are involved. Colors of nodes in PPI networks are representative of FCs in “soft” compared to “hard” plaques.

**Table 1 biomedicines-09-01156-t001:** Main clinical parameters of patients sorted according to the plaque typology (hyperechoic or “hard” plaques and hypoechoic or “soft” plaques).

Parameters	“Hard” (*n* = 31)	“Soft” (*n* = 44)
Age (years) *	67.1 ± 6.9	72.6 ± 8.0
Sex ratio, m/f	22/9	34/10
BMI *	25.3 ± 3.4	25.9 ± 3.5
Triglycerides (mg/dL) *	116.8 ± 48.2	124.1 ± 44.6
Total Cholesterol (mg/dL) *	166.6 ± 41.4	181.3 ± 50.5
HDL Cholesterol (mg/dL) *	47.6 ± 14.1	42.3 ± 15.8
LDL Cholesterol (mg/dL) *	96.8 ± 29.8	112.0 ± 45.0
Cholesterol lowering therapy (%)	23/31 (74.2)	41/44 (93.2)
Glycemia (mg/dL) *	114.4 ± 35.8	107.3 ± 20.9
HbA1C (%) *	6.6 ± 1.0	6.5 ± 1.0
Type 2 Diabetes (%)	9/31 (29.0)	19/44 (43.2)
Glucose lowering Therapy (%)	9/9 (100)	12/19 (63.2)
Systolic blood pressure (mmHg) *	138.8 ± 8.7	129.1 ± 12.6
Diastolic blood pressure (mmHg) *	79.9 ± 8.9	72.4 ± 7.3
Anti-hypertensive Therapy (%)	28/31 (90.3)	35/44 (79.5)

* Values are mean ± SD.

## Data Availability

Data produced in this study are available in the “Appendix A” or may be requested from the corresponding author.

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
