# Peer review of "Apolipoprotein Signature of HDL and LDL from Atherosclerotic Patients in Relation with Carotid Plaque Typology: A Preliminary Report"

_biomedicines, 2021, doi:10.3390/biomedicines9091156_

Round 1
Reviewer 1 Report
The manuscript describes work which appears to have been executed to a high standard and is succinct and of a high standard of presentation. I am happy to recommend publication and only have a couple of very minor comments below.
Line 8 of Abstract. Insert "who" in "...typologies, WHO underwent..."
The same correction is needed in line 2 of the final paragraph of the Introduction.
In the Discussion, the sentence beginning "Besides, salts can be partially substituted by deuterium oxide...". It was not clear what the relevance of this is or whether this is something the authors undertook in their work.
Author Response
We truly appreciate the Reviewer’s positive comments and thank him/her for the suggested corrections (done).
With regard to the comment on the sentence beginning "Besides, salts can be partially substituted by deuterium oxide...", the whole paragraph aims to review, very briefly, the different methodological approaches that have been applied, since the 1950s, to the purification of specific plasma lipoprotein fractions, showing strengths and limitations. The method of ultracentrifugation using low salt concentrations in D2O/sucrose media is, merely, one of them. In our study, we applied a well-established purification method by ultracentrifugation in high-salt media (Lepedda AJ et al. 2013) that allows for the identification of proteins, which have a high probability to be associated with a given lipoprotein fraction.
For the sake of clarity, in the revised version we have specified “In some applications, salts may be partially substituted by deuterium oxide”.
Reviewer 2 Report
Dear Authors
I think that the manuscript contains some new information concerning lipoprotein protein with comparison between patient of hard plague and of soft plague. I had some comments.
Major comments
- Patients were divided by hard and soft plaques with Gray-Weale classification. Authors must write the classification method in detail, and why you used the method, and divided the patients into type 1 and 2, and type 3-5.
- Authors found the different values of some proteins in LDL and HDL between the patients with hard and soft plaques. Authors wrote the discussion of those in page 13, but authors must discuss the result in detail, and add new references.
Minor comments
- Authors must add molecular weight of albumin, ApoB-100, and ApoA1, and show the position of ApoB-100 and ApoA1 by arrow in figure 1 and the legend.
- Authors wrote the limitation in conclusion session. Authors must write that in the last paragraph of discussion session.
Sincerely yours.
Author Response
We thank the Reviewer for the overall positive comments and thank him/her for the suggestions.
Patients were divided by hard and soft plaques with Gray-Weale classification. Authors must write the classification method in detail, and why you used the method, and divided the patients into type 1 and 2, and type 3-5.
According to the reviewer’s suggestion, the classification method has been detailed in “2.1. Sample Collection” section.
Echo-Doppler analysis allows for detecting and characterizing an atherosclerotic plaque and provides an estimate of the stenosis by evaluating flow and the velocities. This method is widely used as a first-line examination in patients with recent cerebrovascular events in order to screen for atherosclerotic disease in peripheral arteries having a number of advantages, including its broad availability, cost-effectiveness, rapidity, absence of ionizing radiation and the possibility of re-examination. Furthermore, it provides a rapid risk stratification of the patient through the Gray–Weale standardization (Huibers A, de Borst GJ, Bulbulia R, Pan H, Halliday A; ACST-1 collaborative group. Plaque Echolucency and the Risk of Ischaemic Stroke in Patients with Asymptomatic Carotid Stenosis Within the First Asymptomatic Carotid Surgery Trial (ACST-1). Eur J Vasc Endovasc Surg. 2016 May;51(5):616-21. doi: 10.1016/j.ejvs.2015.11.013. Epub 2015 Dec 22. PMID: 26725253; Knuuti J, Wijns W, Saraste A, Capodanno D, Barbato E, Funck-Brentano C, Prescott E, Storey RF, Deaton C, Cuisset T, Agewall S, Dickstein K, Edvardsen T, Escaned J, Gersh BJ, Svitil P, Gilard M, Hasdai D, Hatala R, Mahfoud F, Masip J, Muneretto C, Valgimigli M, Achenbach S, Bax JJ; ESC Scientific Document Group. 2019 ESC Guidelines for the diagnosis and management of chronic coronary syndromes. Eur Heart J. 2020 Jan 14;41(3):407-477. doi: 10.1093/eurheartj/ehz425. Erratum in: Eur Heart J. 2020 Nov 21;41(44):4242. PMID: 31504439; Cismaru G, Serban T, Tirpe A. Ultrasound Methods in the Evaluation of Atherosclerosis: From Pathophysiology to Clinic. Biomedicines. 2021 Apr 13;9(4):418. doi: 10.3390/biomedicines9040418. PMID: 33924492; PMCID: PMC8070406).
In our study, patients were sorted, according to the plaque typology, in soft, having a carotid plaque with hypoechoic features (types 1 and 2) and hard, having a carotid plaque with hyperechoic features (types 3, 4, and 5), because there is a plenty of evidence that carotid plaque echolucency provides predictive information in asymptomatic carotid artery stenosis, as reviewed in the meta-analysis study by Gupta et al. (Gupta A, Kesavabhotla K, Baradaran H, Kamel H, Pandya A, Giambrone AE, Wright D, Pain KJ, Mtui EE, Suri JS, Sanelli PC, Mushlin AI. Plaque echolucency and stroke risk in asymptomatic carotid stenosis: a systematic review and meta-analysis. Stroke. 2015 Jan;46(1):91-7. doi: 10.1161/STROKEAHA.114.006091. Epub 2014 Nov 18. PMID: 25406150; PMCID: PMC4280234).
Authors found the different values of some proteins in LDL and HDL between the patients with hard and soft plaques. Authors wrote the discussion of those in page 13, but authors must discuss the result in detail, and add new references.
We partially agree with the reviewer about the importance of discussing the potential involvement of each differentially expressed protein on development of different types of advanced atherosclerotic lesions. By comparing the protein abundances between “soft” and “hard”, a plethora of proteins with |FC| >3 have been identified (86 proteins in HDL and 16 proteins in LDL), which make it very difficult in order to contextualise their functions. Instead, it is probably more informative to analyse the many functional partnerships and interactions that occur among them, which are at the core of the numerous functions of both HDL and LDL. We did so by generating protein-protein interaction networks showing either known or predicted physical/functional protein associations based on genomic context, high-throughput experiments, coexpression and previous knowledge, using STRING v11.0 and by performing Gene Ontology (GO) enrichment analysis, which are ubiquitously used for interpreting high throughput molecular data and generating hypotheses about underlying biological phenomena of experiments (Tomczak A, Mortensen JM, Winnenburg R, Liu C, Alessi DT, Swamy V, Vallania F, Lofgren S, Haynes W, Shah NH, Musen MA, Khatri P. Interpretation of biological experiments changes with evolution of the Gene Ontology and its annotations. Sci Rep. 2018 Mar 23;8(1):5115. doi: 10.1038/s41598-018-23395-2. PMID: 29572502; PMCID: PMC5865181).
Authors must add molecular weight of albumin, ApoB-100, and ApoA1, and show the position of ApoB-100 and ApoA1 by arrow in figure 1 and the legend.
Authors wrote the limitation in conclusion session. Authors must write that in the last paragraph of discussion session.
According to the reviewer’s suggestions:
-we have modified figure 1 by adding arrowhead showing the position of Apo B100 and Apo A1 and the corresponding caption by indicating molecular weights of albumin, Apo B100, and Apo A1;
-we have moved the sentence about the limitation of the study from the conclusion session to the discussion session;
-we have submitted the manuscript to English language revision by a native English-speaker expert in scientific writing (see aknowledgements).